# Evaluating the Effects of Denmark’s New Tobacco Control Act on Young People’s Use of Nicotine Products: A Study Protocol of the §SMOKE Study

**DOI:** 10.3390/ijerph191912782

**Published:** 2022-10-06

**Authors:** Marie Borring Klitgaard, Nanna Schneekloth Jarlstrup, Lisbeth Lund, Anne-Line Brink, Astrid Knudsen, Anne Illemann Christensen, Lotus Sofie Bast

**Affiliations:** 1National Institute of Public Health, University of Southern Denmark, Studiestreade 6, 1455 Copenhagen, Denmark; 2Danish Cancer Society, Strandboulevarden 49, 2100 Copenhagen, Denmark

**Keywords:** tobacco control legislation, youth smoking, youth tobacco use, study protocol, tobacco and nicotine products, POS display ban, plain packaging, tobacco price

## Abstract

(1) Background: In December 2020, a broad majority of political parties in Denmark agreed on a new tobacco control act. In addition, price increases on tobacco in 2020 and 2022 became part of the Danish Finance Act. This study protocol describes the study “§SMOKE–A Study of Tobacco, Behavior, and Regulations” designed to monitor and evaluate the implementation and effect of the new strengthened tobacco control acts. The overall aim is to monitor tobacco use among young people before, during, and after implementation of the new tobacco control legislation, including an increased price on tobacco, a ban on point-of-sale tobacco displays, and plain packaging. Subgoals are to monitor overall use of nicotine products, attitudes, and norms. (2) Methods: This study is designed as a five-year impact evaluation with repeated cross-sectional survey data collections. The baseline survey was conducted before implementing an increased price on tobacco, the first step in the new legislation, initiated 1 April 2020. Study participants (n = 37,500) were a random sample of individuals living in Denmark aged 15 to 29 years. (3) Conclusions: This study examines the impact of the new strengthened tobacco control legislation in Denmark from 2020 to 2025. The findings of this study are relevant to other countries facing implementation of similar measures to explore intended and unintended consequences of the legislation and help to identify how the legislation could be further improved.

## 1. Introduction

Decades of declining cigarette smoking in the adult population in Denmark have been followed by a lengthy period of stagnation [1], and among youth and young adults, smoking prevalence increased from 2013 to 2017 [2]. However, recent surveys show a decrease in smoking prevalence among youth [3,4]. In addition, new tobacco, and nicotine products, i.e., e-cigarettes, heated tobacco, and nicotine pouches have been introduced to the Danish market, thereby creating new patterns of use, i.e., dual use or experimentation with several types of products [5,6].

Many factors influence tobacco use behavior, and tobacco industry marketing, advertising, and promotion are prominent factors encouraging tobacco use [7]. Young people are often a key target group for these tobacco marketing strategies [8]. The best way to prevent smoking and the use of other tobacco and nicotine products among the younger generations is by using a comprehensive approach with a wide range of preventative initiatives [9].

In 2004, Denmark ratified a legally binding treaty, the World Health Organization Framework Convention on Tobacco Control (WHO FCTC). In the preceding decades smoking prevalence has steadily declined thanks to regulatory measures and awareness of the health risks of smoking, and in 2007 the act on smoke-free environments was passed. However, from 2011 and onwards, the decline in smoking prevalence levelled off, and youth smoking was found to rise. In 2016, smoking was estimated to be responsible for 13,600 deaths every year [10]. In recent years, tobacco use among Danish youth has gained significant attention among politicians, with the Danish Government launching a vision of a smoke-free generation in 2016 [11]. Subsequently, the Danish Cancer Society and the TrygFonden foundation initiated the endgame partnership Smoke-Free Future (Roegfri Fremtid) with the aim of no children and a maximum of 5% adults smoking in 2030 [12]. In 2017, the partnership Smoke-free Future initiated a collaboration with WHO Europe and the European Network for Smoking and Tobacco Prevention (ENSP), which identified key challenges and main recommendations for tobacco control in Denmark. These were identified as adopting a comprehensive action plan for tobacco control, decreasing affordability of tobacco, limiting tobacco advertising by implementing plain packaging and a point-of-sale (POS) display ban, and protecting public health policies from the influence of the tobacco industry (1).

Moreover, in 2018, a scientific report on tobacco prevention among Danish youth was published by the Council on Health and Disease Prevention (Vidensråd for Forebyggelse) [13] with recommendations as to which initiatives to prioritize. Furthermore, a Danish study estimating the effect of three different preventive strategies of tobacco prevention among Danish youth pointed to the difficulties of reaching the aims of a smoke-free generation by 2030 [14,15].

### 1.1. The Legislation

In 2019, the Danish parliament proposed implementing a national action plan against smoking among children and young people [16]. The overall objective of the action plan was to reduce smoking prevalence, reduce the attractiveness of tobacco products, reduce exposure to tobacco products, reduce knowledge about tobacco branding, and increase awareness of tobacco harms. The action plan was translated into a series of amendments which were finally adopted in December 2020 and implemented as legislation. Specifically, a series of legislative measures were implemented from 2020 to 2022 (see Table 1). In addition, the Finance Act of 2019 included two tax increases on tobacco, in 2020 and 2022, respectively. Furthermore, as part of the European Tobacco Products Directive, a ban on menthol additives in cigarettes and roll your own (RYO) tobacco was implemented in May 2020.

### 1.2. §SMOKE–A Study of Tobacco, Behavior, and Regulations

The implementation of the new and strengthened tobacco control act poses a unique opportunity to evaluate and monitor the effects of the legislation. Furthermore, it allows for assessing changes in tobacco and nicotine use and how attitudes and norms might change among adolescents and young adults aged 15–29 during and after the implementation period. The §SMOKE study is mainly focused on three parts of the legislation: the increase in tobacco prices, the point-of-sale display ban, and the plain packaging (described below). Other aspects of the legislation are being monitored as well.

Study Aims

The overall aim of §SMOKE—A Study of Tobacco, Behavior, and Regulations is to monitor tobacco use among youth before, during, and after the implementation of increased prices on tobacco, a ban on POS displays, and plain packaging. Subgoals are to monitor the overall use of nicotine products, attitudes, and norms.

#### 1.2.1. Increased Tobacco Prices

Significant increases in the prices of tobacco products have proved to be one of the most cost-effective tools to reduce tobacco use. Evidence shows that price increases on tobacco are highly effective in preventing initiation, encouraging cessation, and reducing consumption among continuing users [17,18]. On average, a 10% price increase in price on a pack of cigarettes reduces demand for cigarettes by about 4% among adults in high-income countries [19]. For youth populations, increased tobacco prices are suggested to have an even greater effect, allowing price interventions to significantly impact youth smoking [17,18]. Compared to other similar countries, tobacco prices in Denmark have been rather low [20]. It can therefore be expected that an increase in price should reduce smoking prevalence and smoking intensity and lead to a transition from daily smoking to occasional or no smoking. Furthermore, an increase in price may heighten the rate of quitting attempts and cessation in youth.

#### 1.2.2. Point-of-Sale (POS) Display Ban

Tobacco promotion and marketing has a major impact on the behavior of children and young adults with respect to tobacco products and may alter their perception of accessibility of tobacco products [21]. In the presence of increasing tobacco advertising bans, POS displays of tobacco in retail establishments enable the tobacco industry to continue advertising products in a way that appeals to young people in particular. Research shows that exposure to tobacco displays at POS increases adolescents’ susceptibility to smoking and smoking initiation [22,23]. Growing evidence shows that a POS display ban is followed by a decrease in adolescents’ perceived tobacco accessibility, a decline in tobacco use, and a more negative attitude towards smoking [9,21,24,25]. A POS display ban was first introduced by Iceland in 2001, followed by Ireland, Thailand, Norway, Scotland, Russia, the UK, and other countries [24,25]. In Denmark, a POS display ban is stipulated to fully cover tobacco products, and such products are only accessible to employees.

Supported by this evidence, it is anticipated that a POS display ban should reduce the perceived accessibility of tobacco products and reduce exposure to tobacco products, which may contribute to a reduction in youth smoking rates.

#### 1.2.3. Plain Packaging

Plain packaging is intended to reduce the promotional appeal of packaging, with the aim of discouraging initiation, reducing tobacco use, and hindering relapse [7,26,27]. Furthermore, standardized packing may alter knowledge, attitudes, and beliefs concerning tobacco use, which might reduce tobacco uptake in children and young people and lead to reduction or cessation of tobacco use, or both, in current tobacco users [26]. Evidence shows that branded packaging is significantly more attractive than plain packaging. On the contrary, plain packaging is thought to be less attractive and to be associated with less positive characteristics among adults and young people. In addition, the removal of brand elements has proven to enhance the salience and effectiveness of health warnings and to reduce misperceptions created by packaging designs techniques that may suggest that certain products are less harmful than others [25,27,28].

## 2. Materials and Methods

### 2.1. Study Design

This study is a nationwide cross-sectional questionnaire-based survey among Danish adolescents and young adults aged 15–29 years. A six-year evaluation was completed with seven subsequent data collections (see Figure 1). The repeated data collections allowed for examination of trends in behavior concerning tobacco and nicotine products and the effects of strengthened tobacco control legislation.

### 2.2. Timing of Data Collection

Data collection for the baseline survey began in January 2020, before implementation of the first step in the strengthened legislation, that is, the increase in tobacco prices. The collection of follow-up data (see Figure 1) was planned to capture reactions to the implementation of specific initiatives during the implementation period.

### 2.3. Study Sample

In Denmark, everyone has a unique personal identification number. This allowed all sample members to be drawn at random from the population using the Danish Civil Registration System. The register contains information on sex, age, address, marital status, citizenship, and place of birth for all individuals with permanent residence in Denmark [29].

The sample for each survey was randomly drawn from among the national population of adolescents and young adults 15–29 years of age. Data on age, gender, ethnicity, regional attachment, municipality attachment, and place of birth were collected and connected to the questionnaire response by a pseudonymized identification number.

The smoking prevalence (daily and occasional) among Danish 15–29-year-olds was 26% at the time of the baseline of this study [30]. To detect changes in smoking prevalence and to allow for subgroup analyses and further stratification, a relatively large number of observations were required. With an expected response rate of 40% in the included age group, the invited number of participants was calculated to be 37,500 individuals. As such, the expected number of respondents was calculated to be approximately 15,000, resulting in approximately 3900 cigarette smokers.

Hence, this sample size allowed us to assess trends by gender, age group, and educational background. Furthermore, the sample size allowed for an examination of trends in the use of tobacco and nicotine products among subgroups with different use statuses, i.e., smokers/users of nicotine products, ever-smokers/users, and never smokers/users. In addition, dual, and multiple use was examined.

### 2.4. Sample for Baseline Survey

The baseline questionnaire was sent to 37,482 individuals. In total, 13,315 returned valid responses, for a response rate of 35.5% (see Figure 2).

### 2.5. Data Collection Procedure

In Denmark, from the age of 15 years everyone is assigned to Digital Post, which is a secured electronical mail service. It is possible to seek an exception to receiving Digital Post due to, for example, not being able to use a computer, which mostly concerns elderly people. Approximately 2% of the Danish population do not receive Digital Post.

An introduction letter was sent to all selected participants through Digital Post with the aim of describing the purpose and content of the survey. It was emphasized that participation was voluntary. Individuals not registered to use Digital Post were sent an invitation and a survey questionnaire through the regular postal service. This procedure was followed only in the baseline survey. In the subsequent surveys, the questionnaires were solely sent through Digital Post, as only a minimal number of questionnaires were returned with regular post. In addition, as the target study population was between the ages 15–29, very few were not registered to use Digital Post.

In all the surveys, two reminders were sent to all invited individuals who had not returned or submitted the questionnaire. The first was sent after 10 to 14 days and the second after the following 10 to 14 days. The data collection period ran for six weeks.

### 2.6. The Questionnaire

The questionnaire consisted of approximately 60 items, and included items addressing different tobacco and nicotine products, topics related to consumer habits and attitudes towards product use, and items related to the implemented tobacco laws. The questionnaire contained sections with items addressing different tobacco and nicotine products, i.e., cigarettes, e-cigarettes, smokeless tobacco (snuff, nicotine pouches, and chewing tobacco), hookahs, and heated tobacco. A core set of items was included in every survey. In addition, new items were added to address emerging themes, such as the COVID-19 pandemic, or items elaborating components of specific tobacco legislation in the follow-up data collection (see themes of the baseline questionnaire in Table 2).

Most of the items had been used in previous Danish health surveys such as the Danish National Health Survey [31,32] and Smoking Habits of Danish citizens [30], or derived from peer-reviewed articles and systematic reviews [9,33,34,35].

### 2.7. Demographic Information from Registers

Information on gender and age was collected through national registers. Gender was stratified by male and female. Based on both the period with the highest risk of starting to smoke cigarettes or use other nicotine containing products and the age group registered for receiving Digital Post, ages 15 until 29 years were included. We constructed three age groups: 15–17 years, 18–24 years, and 25–29 years. This construction allows for studying behavior among, i.e., adolescents not legally allowed to buy tobacco products (15–17-year-olds).

### 2.8. Data Analysis

The primary analysis for data collection was focused on trends in tobacco and nicotine use and the contextual implementation of a legislative initiative. Furthermore, analysis on contemporary events such as the COVID-19 pandemic which may have influenced the data collection and tobacco-related behavior were analyzed.

Specifically, we examined trends related to:(1)The prevalence of smoking and use of specific tobacco and nicotine products between baseline and follow-up surveys in the subsequent years, including dual and multiple product use;(2)Increased tobacco prices, i.e., smoking intensity, transition from daily to occasional smoking or no smoking, quitting attempts and cessation, effects in subgroups;(3)POS display ban, i.e., perceived availability of products, brand awareness, smoking norms, temptation to buy products, and changes in attitudes towards smoking;(4)Plain packaging, i.e., package appeal, noticing packages, covering up packages.

After each data collection, data were quantitatively analyzed in a relevant statistical program such as SAS or Stata. Descriptive statistics were utilized to describe the sample of study participants in each survey. Furthermore, trends over time, χ2-tests, and relevant regression models were used to examine potential differences across survey years.

Findings from each survey were assessed in a broader context by comparing contextual factors that might influence change in outcome and comparison with other data on national as well as international levels.

### 2.9. Weighting

Using the unique personal identification number, both responders and non-responders can be linked on an individual level to different central registers. In addition, by knowing the population distribution of characteristics such as age and gender, these data can be used to assess the respondent sample against population data. Calibrated weights were thereby calculated based on achieving alignment between the sample and the general population data from registers. Hence, by applying calibrated weights, it was to a certain extent possible to statistically allow for differential non-response.

New weights were computed for each sample using additional information from the national register to consider the different sampling probabilities. Weights were computed based on gender and age group.

## 3. Discussion

Implementing such comprehensive national tobacco legislation as that which Denmark has passed calls for thorough examination of the reaction among both tobacco and nicotine product users and non-users. The §SMOKE study provides representative data for Denmark by focusing on youth reactions and behaviors related to the implementation of three selected initiatives in the tobacco legislation: increased tobacco prices, POS display ban, and plain packages.

It is rather unique that such comprehensive tobacco legislation is implemented all at once. Hence, most other studies have examined legislative initiatives separately. Here, we compare the results from the §SMOKE study to relevant findings on the three selected initiatives. Among other things, we expect the increase in price to reduce smoking prevalence and smoking intensity and that it may lead daily smokers to transfer to occasional smoking, as seen previously in [17,18]. The §SMOKE study contributes to the rather limited knowledge on the specific effect of prices on adolescent and youth smoking [36]. A large study on the effect of POS display bans on smoking prevalence found that the ban reduced smoking prevalence by about 7% in the adult population [37]. Studies on youth have shown that a POS ban reduced current smoking, daily smoking, and regular smoking [38], and that adolescents noticed fewer tobacco displays after implementing a POS display ban [39]. The results from the §SMOKE study is compared to, e.g., results from the DISPLAY study conducted in the UK [40] and the evaluation of a POS display ban in Scotland [41]. Plain packaging is expected to reduce the appeal of packs, as seen in, i.e., Canada [42] and the U.S. [43].

The results from the §SMOKE study provide unique data to evaluate the new legislation while it is implemented. Further, it should be possible to follow, e.g., smoking prevalence, use of other products such as e-cigarettes and nicotine pouches, changes between product types and between products of different prices, and other tobacco products that unexpectedly become popular in certain youth groups. With this data, we have a unique opportunity to support the possibility of timely and effective prevention initiatives related to new and not yet known tobacco or nicotine products. These data enable examination of dual and multiple use over time. In this way, the present study can help to assess possible loopholes in the current legislation as well as the response of adolescents to the ongoing market development of nicotine and tobacco products.

Further, the §SMOKE study can help to qualify the next steps of Danish tobacco control on the path towards a smoke-free generation in 2030 as well as contribute to knowledge concerning the effectiveness of various tobacco control measures in an international context.

For future research, the data derived from the §SMOKE surveys can be linked on an individual level to different official statistical registers (e.g., the Danish National Patient Register, the Danish Register of Causes of Death, the Danish National Prescription Register, and the Danish National Service Register) thanks to the unique personal identification numbers, which allow for analyses of the relationship between, e.g., risk factors and morbidity and mortality, social inequality in health, etc. Hence, data from the surveys constitute a unique research database. In this respect, the major strengths are the large number of respondents, the setting in a general population, and the diversity of questionnaire content.

### Methodological Issues

A strength of this study is the use of large sample sizes drawn randomly from a national register. The random sampling procedure and the size of the study populations allow for examinations of behavior, including in sub-groups, as well as multiple types of tobacco or nicotine products. Furthermore, repeated data collection allows for continued monitoring and evaluation of legislation as well as examination of trends.

In the §SMOKE-study, the focus is mainly on three selected areas of the legislation. However, the questionnaire comprises items that can measure other aspects of the legislation, and there is room to adapt the questionnaire to include relevant emerging factors, such as the COVID-19 pandemic, thereby following and reacting to current trends and societal situations. Another strength of the study is the magnitude of items in the questionnaire, covering almost all tobacco and nicotine products, which enables analyses of, e.g., dual, and multiple uses.

Furthermore, we examine differences among participants and non-participants and use weights for age and gender to account for possible bias in the responses due to these factors, thereby enhancing the generalizability of the study results.

Certain general limitations apply to large national surveys such as the §SMOKE study. Youth behaviors are self-reported, incurring the risk of social desirability bias. There is a further risk of reliability biases related to self-reported survey data, as these are based on confidence in the accuracy of the respondents’ recall as well as their motivation to provide truthful information on the topic of interest. However, previous research shows good correspondence between adolescents’ self-reported smoking status and biological measures [44,45]. Furthermore, we regard young people themselves as the best source to explain their own behaviors.

In the baseline survey, the questionnaire was sent by a secured electronical mail service (Digital Post), use of which by individuals living in Denmark is mandatory. However, a small proportion are excepted. These include mainly elderly persons. Furthermore, sample members who were not registered to use Digital Post received a postal letter. Two reminders were sent to the whole sample size to increase the number of respondents.

The response rate of the baseline survey was 35.5%, which is rather low; however, it is in accordance with response rates among youth in other Danish population-based studies. In Denmark, there has been an overall declining trend in participation proportion in health surveys over the past 20–30 years, and the relatively low participation might be a result of questionnaire fatigue among youth, which has been seen in other studies, i.e., the Health and Well-being in High School study from 2019 [46]. To account for the low response rate, we used weights to level out differences according to gender and age. Furthermore, examinations of the residence of respondents showed good concordance between respondents and non-respondents to the survey, thereby strengthening the generalizability of the study results.

## 4. Conclusions

The §SMOKE study can provide real-world data on the prevalence of tobacco use before, during, and after implementation of increased prices of tobacco products, a POS display ban, and plain packaging. It thus complements the growing global evidence on the effect of such national tools to reduce tobacco use. The results are of considerable interest to, e.g., policymakers and can help to pinpoint successes due to the law as well as specific areas that could benefit from further restrictions.

In addition, this study can provide an opportunity to examine the way in which exceptions to the act lead to a shift towards tobacco and nicotine products that are less strictly regulated, e.g., with respect to plain packaging or characterization of flavors.

Future authors should discuss the results and how they can be interpreted from the perspective of previous studies and their working hypotheses. These findings and their implications should be discussed in the broadest context possible. Future research directions should be highlighted.

## Figures and Tables

**Figure 1 ijerph-19-12782-f001:**
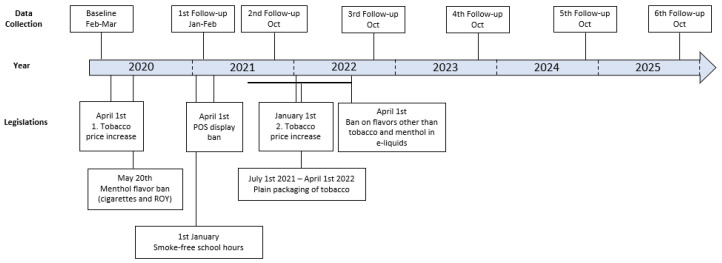
Timeline: an overview of data collection and implementation of laws.

**Figure 2 ijerph-19-12782-f002:**
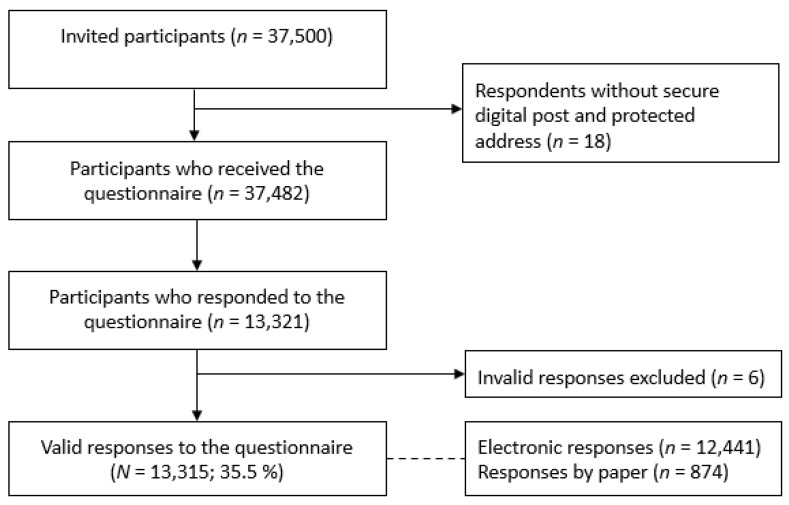
Flowchart of §SMOKE data collection–Baseline 2020.

**Table 1 ijerph-19-12782-t001:** Overview of the most important legislations implemented from 2020–2022.

Legislations	Details	Came into Effect
First increase in tobacco prices	The average price of a pack (20 PCS) of cigarettes increased from 40 DKK/5.35 EUR to 55 DKK/7.36 EUR (37.5% increase)	1 April 2020
Menthol flavor ban (Tobacco Products Directive)	Ban on cigarettes and RYO tobacco with a characterizing flavor of menthol	20 May 2020
Smoke-free school hours	No smoking during the school day in primary school, even outside school premises	1 January 2021
Point-of-sale display ban	No display of tobacco products, herbal products, e-cigarettes, or nicotine products at points of sale in retail outlets	1 April 2021
Ban on flavors other than tobacco and menthol	A proposed ban on flavors other than tobacco and menthol in other tobacco products than cigarettes and RYO (e.g., chewing tobacco, heated tobacco, and shisha/waterpipe) was planned, but this currently awaits a decision by the European Commission	*Temporarily suspended*
Smoke-free-school-hours	No smoking during school hours in high schools and vocational schools, even outside school premises	31 July 2021
Second increase in tobacco prices	Increase in the average price of a pack (20 PCS) of cigarettes from DKK 55/EUR 7.36 to DKK 60/EUR 8.02 (9% increase)	1 January 2022
Ban on flavors other than tobacco and menthol in e-liquids	A ban on flavors other than tobacco and menthol in e-liquids used in e-cigarettes	1 April 2022
Plain packaging of all forms of tobacco including cigarettes, RYO tobacco, chewing tobacco, heated tobacco and waterpipe tobacco	Transition period: 1 July 2021 until 1 April	1 April 2022
Plain packaging of e-cigarettes	Transition period: 1 October 2021 until 1 October 2022	1 October 2022
Definition of tobacco surrogates, including nicotine containing products which are not medically approved for smoking cessation. Tobacco surrogates are in general covered by the same legislation as tobacco products except standardized packaging and the suggested flavor ban. This includes nicotine pouches		1 July 2022

**Table 2 ijerph-19-12782-t002:** Overview of the §SMOKE questionnaire and themes (baseline questionnaire).

Topics	Indicators
Sociodemographics	Gender, age, highest level of educational attainment, living status (alone, with parents, etc.)
Quality of life	Quality of life, physical and mental health, and well-being
Cigarette use	Current smoking status, smoking frequency and quantity, age at regularly smoking, flavor additives, quit attempts and intentions, intention to smoke (again), perception of smoking as acceptable, smoking in the family and among relatives
Other tobacco and nicotine products	Products: E-cigarettes, smoke-free tobacco (e.g., nicotine pouches), water pipe, cigarillos, heated tobaccoTobacco and nicotine use–frequency and type of product, flavor additives, perception of health risk, quit attempts and intentions
Increased cigarette prices	Where to buy cigarettes, purchase price, factory-made or ROY cigarettes, smokers self-assessed effect of increased price on frequency of smoking, financial stress (shortage of money)
POS display ban	Notification of visible tobacco products in retail establishments, social media, movies, or the internet in the past month
Plain packaging	Brand awareness, perceptions of quality, satisfaction, taste, value for money, harmfulness, the appeal of the package, risk for addiction and type of cigarette (‘light’, slim, organic, etc.), concerns about health effects, notification of health warnings

## Data Availability

The data collected in this study are not publicly available. However, the dataset used and/or analyzed during the current study is available from the corresponding author(s) on responsible request.

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
