# Peer review of "Evaluating the Effects of Denmark’s New Tobacco Control Act on Young People’s Use of Nicotine Products: A Study Protocol of the §SMOKE Study"

_ijerph, 2022, doi:10.3390/ijerph191912782_

Round 1
Reviewer 1 Report
This is an important public health issues. The paper presents the methods for a national survey of 15 to 29-year-olds. The presentation is excellent, and the details are generally clear.
Two suggestions to improve the paper for an international audience are:
1) Explain the digital post system. How are children registered to it as the cohort to be surveyed includes children aged under 18 years? How does it work - is it via an email system. This information would be helpful for international readers who would be unfamiliar with the process.
2) What international comparisons can be included? What similar surveys - even in part - are undertaken? This would inform readers and put this research and Denmark in context.
Many thanks
Author Response
Dear Reviewer,
Thank you for constructive comments on our manuscript.
Please find point-by-point responses below. We hope that you will find our corrections relevant and satisfying.
Reviewer 1:
This is an important public health issues. The paper presents the methods for a national survey of 15 to 29-year-olds. The presentation is excellent, and the details are generally clear.
Two suggestions to improve the paper for an international audience are:
- Explain the digital post system. How are children registered to it as the cohort to be surveyed includes children aged under 18 years? How does it work - is it via an email system. This information would be helpful for international readers who would be unfamiliar with the process.
RESPONSE: Thank you for this question. The Digital Post system is registered by the individual personal registration number which is a unique number that everyone in Denmark have. From age 15, you are assigned to Digital Post, unless you seek for exception due to i.e., not being able to use a computer. Al official mail is sent through your digital post, e.g., messages from the municipality and messages about taxes. We added an explanation of this in the manuscript on page 5.
- What international comparisons can be included? What similar surveys - even in part - are undertaken? This would inform readers and put this research and Denmark in context.
RESPONSE: Overall, there has been quite a lot of focus on tobacco prevention during the past decades in many countries. Most places the legislation has been implemented one initiative at the time. For example, POS (point of sales) display ban was first introduced by Island in 2001, followed by Ireland, Thailand, Norway and Scotland, Russia, UK, and other countries – this is described underneath the three legislations that the §SMOKE study focuses on. Standardized tobacco packages have also been introduced in many countries and there is quite a lot of studies about tobacco taxation, however not so much about the effect among youth in European/high income countries. We have added to the discussion about this.
Editor comment:
"Dear Authors,
I wondered upon reading the paper why you did not include at least some baseline results?
Please address this question after peer review in your response to the reviewers. "
RESPONSE: I can see why you posed this question. And we also talked about doing that. However, explaining the study design in sufficient details were the primary focus, and we did not feel that we could present both the study and evaluation design and enough baseline results to give a clear overview in the same paper. There’s multiple important outcome measure to report in this study.
Baseline results are previously published in Bast et al. 2022: “Use of Tobacco and Nicotine Products among Young People in Denmark—Status in Single and Dual Use” Int. J. Environ. Res. Public Health 2022, 19, 5623. https://doi.org/10.3390/ijerph19095623.
Reviewer 2 Report
Dear Klitgaard et al.
I was afforded the opportunity to review your manuscript entitled "Evaluation of implementation of increased tobacco price, tobacco point-of-sale display ban and plain packaging in Denmark: A study protocol for monitoring the impact on 15-29-year-olds tobacco use, 202-2025" .
I want to congratulate you on a well-prepared manuscript. A few comments and suggestions for your consideration:
The title is quite long. I would consider shortening it to, as an example: "Evaluating the effects of Denmark's new tobacco control act on young people's use of nicotine products: A study protocol". This is merely a suggestion.
Throughout the manuscript there are several typographical errors (double spaces in line 38, " First" in Table 1, Row 1 as an example). Although the editorial team will fix most of these, it is good practice to sort out as many of these prior to the final submission.
Please include the year in all dates. This is especially needed in Table 1.
Line 200: "approximately 2% of Denmark's population". The % sign should always be without a space. Also, ensure that the reader knows it is 2% of Denmark's population.
Line 230-231: Why were these age groups chosen? Motivate the choice.
Line 337: Refers to other studies, but only cites a single study.
Double check references for consistency in spacing and format according to IJERPH guidelines. Also include the DOI for scientific works.
I wish you well with the study, and know that it will bring new insights into the nicotine use habits of young people.
Kind regards
Author Response
Dear Reviewer, thank you for the positive comments. Please find point-by-point to your requests below.
Reviewer comments:
I want to congratulate you on a well-prepared manuscript. A few comments and suggestions for your consideration:
The title is quite long. I would consider shortening it to, as an example: "Evaluating the effects of Denmark's new tobacco control act on young people's use of nicotine products: A study protocol". This is merely a suggestion.
RESPONSE: Thank you for the positive comments and this suggestion. We changed the title according to your suggestion and added the name of the study to the title.
Throughout the manuscript there are several typographical errors (double spaces in line 38, " First" in Table 1, Row 1 as an example). Although the editorial team will fix most of these, it is good practice to sort out as many of these prior to the final submission.
RESPONSE: We have read and corrected these.
Please include the year in all dates. This is especially needed in Table 1.
RESPONSE: Done.
Line 200: "approximately 2% of Denmark's population". The % sign should always be without a space. Also, ensure that the reader knows it is 2% of Denmark's population.
RESPONSE: As requested by the other reviewer, we have rewritten the section about Digital Post (the data collection procedure), so that it is clearer. We think that the new text also clarifies the issue about the 2%.
Line 230-231: Why were these age groups chosen? Motivate the choice.
RESPONSE: These age groups were chosen based on several issues; age groups in comparable studies, age groups with the highest use of cigarettes and other nicotine containing products, and also based on the recruiting strategy (using Digital Post which applies for all persons aged 15 years or more living in Denmark). We have added at little about this in the text too.
Line 337: Refers to other studies, but only cites a single study.
RESPONSE: We corrected this.
Double check references for consistency in spacing and format according to IJERPH guidelines. Also include the DOI for scientific works.
RESPONSE: We ran through the reference list as requested.
I wish you well with the study, and know that it will bring new insights into the nicotine use habits of young people.
RESPONSE: Thank you very much.
Editor comment:
"Dear Authors,
I wondered upon reading the paper why you did not include at least some baseline results? Please address this question after peer review in your response to the reviewers. "
RESPONSE: I can see why you posed this question. And we also talked about doing that. However, explaining the study design in sufficient details were the primary focus, and we did not feel that we could present both the study and evaluation design and enough baseline results to give a clear overview in the same paper. There’s multiple important outcome measure to report in this study.